# Characteristics of Diaphragmatic and Chest Wall Motion in People with Normal Pulmonary Function: A Study with Free-Breathing Dynamic MRI

**DOI:** 10.3390/jcm11247276

**Published:** 2022-12-08

**Authors:** Xiaoyan Yang, Haishuang Sun, Mei Deng, Yicong Chen, Chen Li, Pengxin Yu, Rongguo Zhang, Min Liu, Huaping Dai, Chen Wang

**Affiliations:** 1Capital Medical University, Beijing 100069, China; 2National Center for Respiratory Medicine, National Clinical Research Center for Respiratory Diseases, Institute of Respiratory Medicine, Chinese Academy of Medical Sciences, Department of Pulmonary and Critical Care Medicine, Center of Respiratory Medicine, China-Japan Friendship Hospital, Beijing 100029, China; 3Department of Respiratory Medicine, The First Hospital of Jilin University, Changchun 130021, China; 4Department of Radiology, China-Japan Friendship Hospital, Beijing 100029, China; 5Institute of Advanced Research, Infervision Medical Technology Co., Ltd., Beijing 100025, China

**Keywords:** magnetic resonance imaging, respiratory muscle, chest wall, diaphragm, respiratory function

## Abstract

Objective: We aimed to quantitatively study the characteristic of diaphragm and chest wall motion using free-breathing dynamic magnetic resonance imaging (D-MRI) in Chinese people with normal lung function. Methods: 74 male subjects (mean age, 37 ± 11 years old) were prospectively enrolled, and they underwent high-resolution CT(HRCT), pulmonary functional tests (PFTs), and D-MRI in the same day. D-MRI was acquired with a gradient-echo sequence during the quiet and deep breathing. The motion of the diaphragm and chest wall were respectively assessed by measuring thoracic anteroposterior diameter (AP), left–right diameter (LR), cranial–caudal diameter (CC), and thoracic area ratios between end-inspiration and end-expiration. The effect of age, body mass index (BMI), and smoking on respiratory muscle function was also analyzed. Results: The mean ratio of right and left AP was greater than that of LR on three transversal planes during both quiet and deep breathing. The mean ratio at the anterior diaphragm (AND, Quiet: 1.04 ± 0.03; Deep: 1.15 ± 0.09) was weaker than that of the apex (vs. APD, Quiet: 1.08 ± 0.05, *p* < 0.001; Deep: 1.29 ± 0.12, *p* < 0.001) and posterior diaphragm (vs. POD, Quiet: 1.09 ± 0.04, *p* < 0.001; Deep: 1.30 ± 0.12, *p* < 0.001) both in quiet and deep breathing. Compared with non-smokers, the left AP and thoracic area ratios in smokers were significantly decreased (*p* < 0.05). However, the ratios of AP, LR, CC, and thoracic area on each plane were similar among groups in different age and BMI. Conclusions: During both quiet and deep breathing, the chest wall motion is prominent in the anteroposterior direction. The motions of diaphragm apex and posterior diaphragm were more prominent than that of the anterior diaphragm. Smoking may affect the respiratory muscle mobility. Dynamic MRI can quantitatively evaluate the motion of respiratory muscles.

## 1. Introduction

Diaphragm and chest wall muscles are the main respiratory muscles and play a very important role in respiratory function. The elevation of respiratory muscle function is vital for diseases that interfere with diaphragm and chest wall dysfunction [1,2,3,4]. Currently, the function of respiratory muscle can be assessed with pulmonary function tests (PFTs), computed tomography (CT), ultrasonography, and transdiaphragmatic pressure [5]. However, there are some limitations to these methods, such as invasive, ionizing radiation, and the inability to measure the chest wall and diaphragm motion synchronously.

Free-breathing Dynamic Magnetic Resonance Imaging (D-MRI), a non-invasive radiation-free method, can be accurately used to measure the lung volume, chest wall, and diaphragmatic motions with the application of multiplanar gradient-echo sequence during free breathing. D-MRI can be used to assess the motion of diaphragm and chest wall in healthy cohort [1,6,7,8,9] and some diseases with respiratory muscle dysfunction, such as chronic obstructive pulmonary disease (COPD) [10,11,12], adolescent idiopathic scoliosis (AIS) [13,14,15], and neuromuscular diseases [16,17,18,19]. These indicate that D-MRI is not only can be used to evaluate the severity of disease but also can be applied to assess the treatment efficacy. However, the characteristics of diaphragmatic and chest wall motion in thoracic anterior-posterior (AP), left-right (LR), and cranial-caudal (CC) directions using D-MRI was limited. In addition, the effect of smoking, age, and BMI on chest wall and diaphragm motion also remains unknown. We aimed to study the characteristics of diaphragmatic and chest wall motion of subjects with normal lung function using D-MRI in the thoracic anterior-posterior (AP), left-right (LR), and cranial-caudal (CC) directions and analyze the effect of age, BMI, and the tobacco smoke exposure on diaphragm and chest wall motion.

## 2. Materials and Methods

### 2.1. Study Cohort

This study was approved by our Institutional ethics committee (2022-KY-031), and all participants provided written informed consent. From October 2021 to June 2022, a total of 74 male subjects (mean age, 37 ± 11 years old) with normal high-resolution CT (HRCT) findings and normal PFTs were prospectively enrolled. All participants underwent HRCT, PFTs, and chest MRI in the same day. The subjects with hypertension, diabetes, scoliosis, congenital heart disease, pacemaker and coronary stent, neuromuscular diseases, and tumor were all excluded.

### 2.2. Pulmonary Function Tests

All subjects underwent PFTs (MasterScreen, Vyaire Medical GmbH, Hoechberg, Germany) at upright seated according to the standards of ATS/ERS [20]. The percentage of predicted forced vital capacity (FVC%), percentage of forced expiratory volume in one second (FEV1%), FEV1/FVC%, percentage of predicted total lung capacity (TLC%), and percentage of predicted DLco corrected for measured hemoglobin (DLco%) were measured.

### 2.3. MRI Protocol

All subjects underwent chest MRI on a 1.5T MRI scanner (MAGNETOM Aera; Siemens Healthcare, Erlangen, Germany) with an 18-channel phased-array body coil and 12-channel spine coil. D-MRI was acquired with a dynamic fast spoiled gradient-recalled echo sequence with parallel imaging acceleration during quiet and deep free breathing in the following key parameters: TR = 868 ms; TE = 46 ms; flip angle = 160°; FOV = 340 mm × 340 mm; resolution = 0.8 mm × 0.8 mm × 10.0 mm; PAT = 2. A total of 40 frames were acquired during the time of 16 s at each plane. The images were respectively obtained at the coronal, sagittal, and transversal planes [1,19]. Figure 1 shows the transversal images respectively acquired at the aortic arch level (Figure 1a), tracheal carina level (Figure 1b), and liver apex level (Figure 1c). The coronal plane was obtained through tracheal bifurcation (Figure 1d), and a right sagittal plane passed through the lung apex and diaphragm apex (Figure 1e). Before MRI scan, all subjects were instructed to practice the quiet breathing and deep breathing.

### 2.4. MR Image Processing and Measurements

Chest D-MR images processing was performed with InferScholar software (https://www.infer-vision.com/) (accessed on 20 May 2022). The details of image processing and quantitative analysis are summarized in Figure 2, including four steps: Step 1 (Figure 2a). MR image segmentation. Two chest radiologists respective with 20 years and 7 years of experience manually segmented images in the end-expiration and end-inspiration on D-MRI. In order to assess the inter-observer agreement of image annotation, 10 MR images were randomly selected to be annotated by one radiologist. Step 2 (Figure 2b). Marking referenced points. The referenced points were marked in the anterior–posterior (AP), left–right (LR), and cranial–caudal (CC) directions along the inner margin of thorax [1,19]. To confirm that the reference points were correctly indicated, all points were checked by two radiologists together.

Step 3 (Figure 2c,d). Measurements extraction and definition. The thoracic area was also extracted from each plane along the inner margin of chest wall. The length was directly extracted in the anterior–posterior (AP), left–right (LR), and cranial–caudal (CC) directions according to the points along the outline of lung [1,19]. The motion of chest wall was elevated by the anterior–posterior distance (AP) and left–right distance (LR) at three transversal planes including the aortic arch, tracheal carina, and the liver apex levels (Figure 2b). The AP distance is defined as the longest distance from dorsal to ventral of the lung (Figure 2c) and the LR distance is defined as the longest distance from the right inner edge to the left inner margin of thorax. The motion of diaphragm was elevated by the cranial–caudal (CC) distance measured as the distance between the lung apex and the anterior, apex, and posterior diaphragm on coronal and sagittal planes (Figure 2d).

Step 4. Measurements analysis. The mean values of end-expiration and end-inspiration among 2–4 respiratory cycles were used to further analysis (Appendix A). We calculated AP, LR, CC, and thoracic area ratios by dividing end-inspiration outcomes by end-expiration outcomes.

### 2.5. Statistical Analysis

All statistical analyses were performed with SPSS 26.0 (IBM Corp, Armonk, NY, USA). Numerical data were present as mean ± SD, while categorical data were present as numbers (percentage). The Mann–Whitney test and Kruskal–Wallis test were used for comparisons. The intraclass coefficients (ICC) were used to determine the inter-observer agreement. ICCs were classified from null (=0) to very good (>0.80) and almost perfect (>0.95) [21]. The significance level was set at *p* ≤ 0.05.

## 3. Results

### 3.1. The Demographic Characteristics

A total of 74 male subjects (mean age: 37 ± 11 years, age range: 22–67 years) prospectively enrolled in this study. Twenty-four males were non-smokers, and the other 50 males were smokers. The mean BMI was 24.2 ± 2.9 kg/M^2^. The metrics of PFTs including mean FVC, FEV1, FEV1/FVC, TLC, and DLco, respectively, were 102.6 ± 10.7, 98.1 ± 11.0, 82.9 ± 8.7, 97.1 ± 8.3, and 100.2 ± 14. 3 in the predicted normal range. The demographic characteristics of all subjects is shown in Table 1.

### 3.2. Quantitative Analysis of Chest Wall and Diaphragmatic Motion

The mean ratios of right and left AP were higher than the LR ratios on three transversal planes in both quiet breathing (Figure 3) and deep breathing (Figure 4). The mean ratio of the distance between the lung apex and anterior diaphragm on end-expiration and end-inspiration (AND, Quiet: 1.04 ± 0.03; Deep: 1.15 ± 0.09) was lower than the mean ratio of the distance between the lung apex and diaphragm apex on end-expiration and end-inspiration (vs. APD, Quiet: 1.08 ± 0.05, *p* < 0.001; Deep: 1.29 ± 0.12, *p* < 0.001) and the distance between the lung apex and posterior diaphragm on end-expiration and end-inspiration (vs. POD, Quiet: 1.09 ± 0.04, *p* < 0.001; Deep: 1.30 ± 0.12, *p* < 0.001) on the right sagittal plane on both quiet breathing (Figure 3d) and deep breathing (Figure 4d). However, the mean ratio was comparable between APD and POD (Quiet: 1.08 ± 0.05 vs. 1.09 ± 0.04, *p* = 0.142; Deep: 1.29 ± 0.12 vs. 1.30 ± 0.12, *p* = 0.344). Figure 5 shows the chest wall and diaphragmatic motion metrics on coronal plane and sagittal plane in end-inspiration and end-expiration. The inter-observer agreement between manual measurements was very good (Quiet breathing, ICC = 0.86; Deep breathing, ICC = 0.91).

### 3.3. Theeffect of Age and BMI on Chest Wall and Diaphragm Motion

As shown in Table 2, the ratios of AP, LR, CC, and thoracic area were comparable on each plane among different age groups, except the thoracic area ratio in the liver apex level during quiet breathing. The mean ratio of the thoracic area in the older people (>50 years old) (1.05 ± 0.06) was more than in the other 2 younger groups (age < 30 years old: 1.02 ± 0.02, 30–50 years old: 1.01 ± 0.03, *p* = 0.001) during quiet breathing. Table 3 indicated that the AP, LR, CC, and thoracic area ratios in each plane had no statistically significant differences between groups in BMI < 25 kg/m^2^ and group in BMI ≥ 25 kg/m^2^.

### 3.4. The Influence of Smoking on Chest Wall and Diaphragm Motion

During quiet breathing, Table 4 showed that the ratio of LAP and thoracic area on the liver apex level had significant differences between smokers and non-smokers (LAP ratios, 1.20 ± 0.03 vs. 1.40 ± 0.04, *p* = 0.024; thoracic area ratio, 1.02 ± 0.05 vs. 1.03 ± 0.03, *p* = 0.034). During deep breathing, the thoracic area ratios on the tracheal carina level in smokers were lower than in non-smokers on the tracheal carina level (non-smoker vs. smoker: 1.42 ± 0.18 vs. 1.33 ± 0.12, *p* = 0.016) and the coronal plane (non-smoker vs. smoker: 1.93 ± 0.28 vs. 1.69 ± 0.31, *p* = 0.003), respectively (Table 4). In addition, the ratio of AP, LR, CC, and thoracic area had no significant differences on other planes among non-smokers and smokers in both quiet breathing (*p* > 0.05) and deep breathing (*p* > 0.05).

## 4. Discussion

In this study, using 2D dynamic MRI, we explored the characteristics of chest wall and diaphragmatic motion in three vertical directions (thoracic anterior–posterior, left–right, cranial–caudal direction) during quiet and deep breathing in males with normal lung function. We found that the motion of chest wall is more prominent in the anterior-posterior direction on three transversal planes representing the upper, middle, and lower chest wall during both quiet and deep breathing. In addition, the motion of the diaphragm apex and posterior diaphragm was greater than that of the anterior diaphragm in the sagittal plane in both quiet and deep breathing, but there are similarities between diaphragm apex and posterior diaphragm. Moreover, both the chest wall and diaphragmatic motion are influenced by smoking but not by age and BMI.

Dynamic MRI has been used to detect early signs of diaphragmatic weakness in patients with Pompe disease [16,17,19,22] and Duchenne muscular dystrophy (DMD) [18,23]. The detailed motions of chest wall and diaphragm in different directions are helpful for us to evaluate the subtle influence of diseases on respiratory muscles. In the present study, we showed that the motion of chest wall in the anterior–posterior direction was greater than in the left–right direction in the transversal planes both in quiet and deep breathing, which is consistent with Groote et al. [7]. This suggests that the motion of the chest wall during inspiration is the greatest in the dorsal–ventral direction in bilateral lung. In addition, the chest wall motion in the anteroposterior direction between right and left sides was comparable at three transversal planes both in quiet and deep breathing, which suggests that the right and left sides of the thorax are a single function unit during breathing.

The diaphragm is the main respiratory muscle and plays an important role in ventilation [24]. Consistent with previous studies [1,25], we found that the motion of the anterior diaphragm was weaker than that of the apex and posterior diaphragm both in quiet and deep breathing. This suggested that the contribution of diaphragm apex and posterior diaphragm to ventilation was greater than that of the anterior diaphragm. Moreover, the diaphragm did not uniformly move during breathing, and it did not act as a single functional unit during active breathing. Furthermore, our study showed that the motion was no different between apex and posterior diaphragm. However, Takazakura et al. found the movement of the diaphragm in the middle was weaker than in the posterior during deep breathing [25]. Traser et al. also demonstrated that the degree of diaphragm excursion decreased significantly from posterior to middle during breathing and phonation [8]. This may be related to the different measurement points we selected.

It has been reported that the aging-related decline in skeletal muscle mass and strength begins at 30 years of age and may correlate with hospitalization and mortality due to some chronic diseases [26]. However, we found the chest wall and diaphragmatic motion were similar among subjects of different ages. This result was consistent with kondo et al. [27]. They used MRI to evaluate the motion of chest wall and diaphragm among healthy young and elder subjects. In contrast, some studies reported that the diaphragmatic function using ultrasonography is significantly weaker in individuals below 30 years when compared with those aged more than 30 years during quiet and deep breathing [28,29]. We thought that different parameters measured using ultrasonography and MRI might cause the different between those investigations. Consistent with Kabil et al. [29], for motion of chest wall and diaphragm, we did not find a significant statistical difference between males with BMI < 25 kg/m^2^ and BMI ≥ 25 kg/m^2^. Kantarci et al. [28] showed a significant difference in diaphragmatic motion between BMI groups and explained that this may be related to the difference of fat and muscle composition. It has been demonstrated that smoking has an important extrapulmonary toxicity including systemic inflammation and muscle dysfunction [30,31,32,33]. In the current study, we found the motions of chest wall and diaphragm in dorsal–ventral and cranial–caudal directions were weaker in smokers, in comparison with non-smokers, which further proved that the respiratory muscle function is also influenced by smoking, although the detailed mechanism remains unknown.

According to our initial results, we supplied the characteristics of chest wall and diaphragm motion in different directions in the normal lung function of males using D-MRI. Compared with PFTs, D-MRI can help clinicians know the regional or subtle alteration of respiratory muscle function under disease conditions, explaining some clinically relevant pathophysiological phenomena, e.g., the cranial–caudal movement of the diaphragm is reduced in patients with emphysema while the dorsal–ventral movement is reduced in patients with IPF [34], and also help to quantitatively analyze the severity of diseases. However, there are several limitations to this study. First, only a small number of male subjects with normal lung function at a single center were included in this research which indicates lack of generalizability. Second, although smoking affects the motion of chest wall and diaphragm, the effects of smoking duration and amounts and types of tobacco on the motion of the respiratory muscles are still unclear. Third, Multiplanar 2D dynamic MRI was used to quantitatively study the motion of chest wall and diaphragm which required a longer acquisition time. In the future, the 3D dynamic MRI will make it easier to assess the characteristics of chest wall and diaphragm motions simultaneously.

## 5. Conclusions

The motion of chest wall is prominent in the anterior–posterior direction during quiet and deep breathing. Meanwhile, the motions of diaphragm in apex and posterior diaphragm on a sagittal plane are stronger than in the anterior diaphragm both in quiet and deep breathing. Smoking affects respiratory muscle mobility. Dynamic MRI can quantitatively evaluate the motion of respiratory muscles.

## Figures and Tables

**Figure 1 jcm-11-07276-f001:**
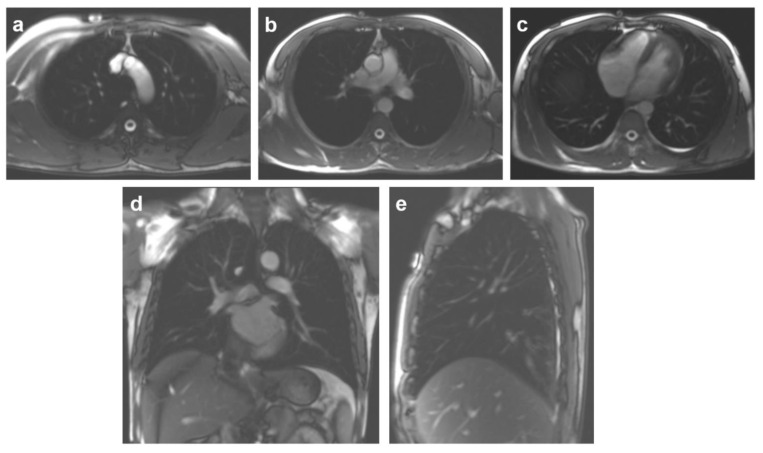
The multiplanar images were acquired on dynamic MRI. The transversal planes including the aortic arch (**a**), tracheal carina (**b**), and the liver apex levels (**c**), a coronal plane through tracheal bifurcation (**d**), and a right sagittal plane passing through the lung apex and diaphragm apex (**e**).

**Figure 2 jcm-11-07276-f002:**
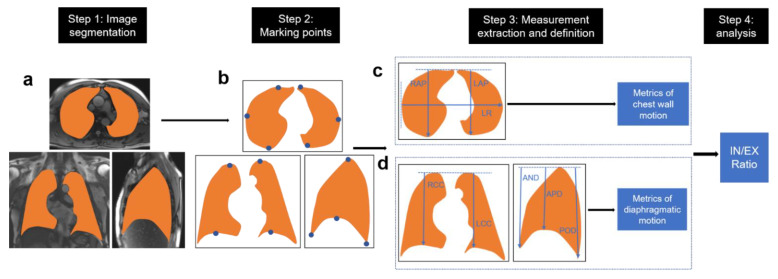
The flowchart of MR image processing and Measurements. MR image segmentation (**a**). Marking referenced points (**b**). Measurements extraction and definition (**c**,**d**). RAP, the anterior–posterior distance of right lung on the transversal plane; LAP, anterior–posterior (AP) distance of left lung on the transversal plane; LR, the left–right distance on the transversal plane; RCC, the cranial–caudal distance of right lung on coronal plane; LCC, the cranial–caudal distance of left lung on coronal plane; AND, the distance between the lung apex and anterior diaphragm on sagittal plane; APD, the distance between the lung apex and diaphragm apex on sagittal plane; POD, the distance between the lung apex and posterior diaphragm on sagittal plane. IN, inspiration. EX, expiration.

**Figure 3 jcm-11-07276-f003:**
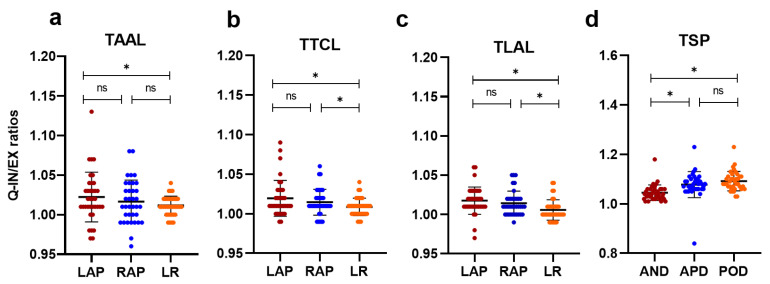
The scatter plots present the difference of ratios between different directions on each plane during quiet breathing (**a**–**d**). Q-IN/EX ratios, the ratio between end-inspiration and end-expiration in quiet breathing; TAAL, the aortic arch level; TTCL, the tracheal carina level; TLAL, the liver apex level; TSP, The right sagittal plane; LAP, the anterior-posterior distance of left thorax; RAP, the anterior-posterior distance of right thorax; LR, the left to right distance of thorax; AND, the distance between the lung apex and anterior diaphragm on sagittal plane; APD, the distance between the lung apex and diaphragm apex on sagittal plane; POD, the distance between the lung apex and posterior diaphragm on sagittal plane. ns indicated *p* > 0.05, * indicated *p* < 0.05.

**Figure 4 jcm-11-07276-f004:**
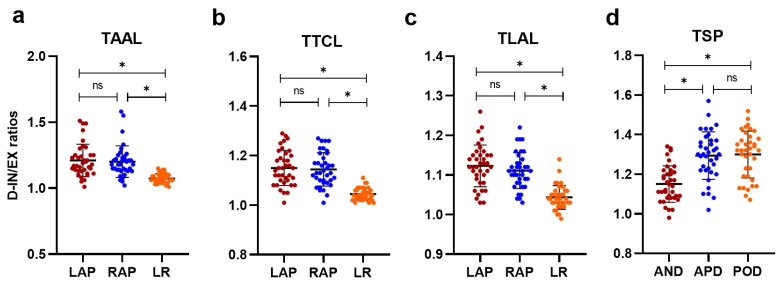
The scatter plots present the difference of ratios between different directions on each plane during deep breathing (**a**–**d**). D-IN/EX ratios, the ratio between end-inspiration and end-expiration in deep breathing; TAAL, the aortic arch level; TTCL, the tracheal carina level; TLAL, the liver apex level; TSP, the right sagittal plane; LAP, the anterior–posterior distance of left thorax; RAP, the anterior–posterior distance of right thorax; LR, the left to right distance of thorax; AND, the distance between the lung apex and anterior diaphragm on the sagittal plane; APD, the distance between the lung apex and diaphragm apex on sagittal plane; POD, the distance between the lung apex and posterior diaphragm on sagittal plane. ns indicated *p* > 0.05, * indicated *p* < 0.05.

**Figure 5 jcm-11-07276-f005:**
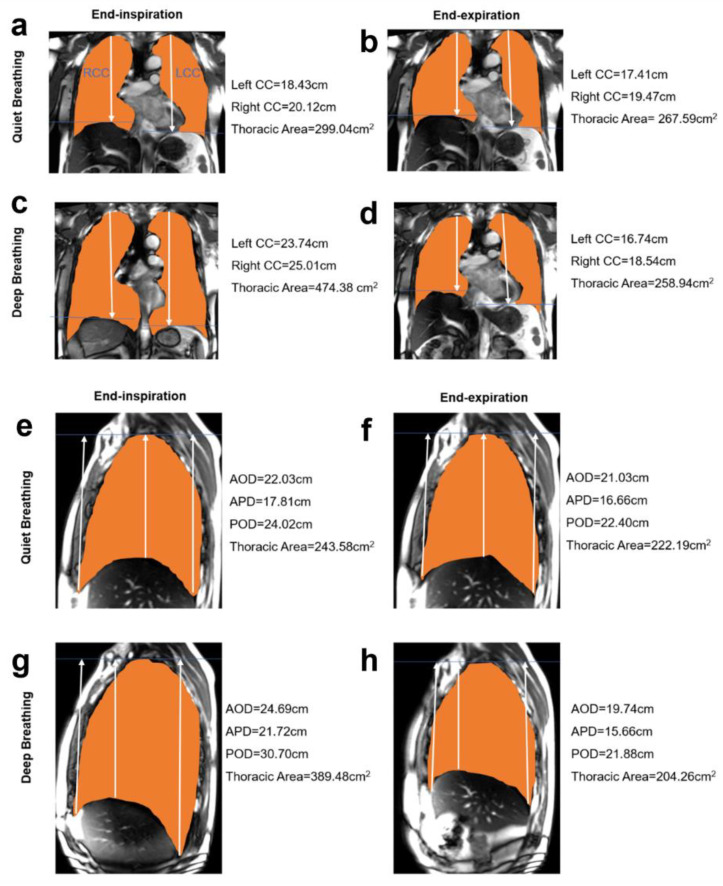
Chest wall and diaphragmatic motion metrics on the coronal plane and sagittal plane in a 50-year-old non-smoker. (**a**–**d**) The right, left CC, and thoracic area on the coronal plane in end-inspiration and end-expiration. (**e**–**h**) The AOD, APD, POD, and thoracic area on the sagittal plane in end-inspiration and end-expiration.

**Table 1 jcm-11-07276-t001:** The characteristics of included subjects.

Subjects	Clinical Characteristics
Number	74
Gender	Male
Mean age (years) (range)	37 ± 11 (22–67)
Height (cm)	171 ± 5.7
Weight (kg)	70.4 ± 7.7
BMI (kg/m^2^)	24.2 ± 2.9
Smoker, n (%)	24 (32.4%)
Smoking habits (pack/years)	4–50
Duration of smoking (years)	2–35
Pulmonary Function Tests	
FEV1% predicted	98.1 ± 11.0
FVC % predicted	102.6 ± 10.7
FEV1/FVC % predicted	82.9 ± 8.7
TLC % predicted	97.1 ± 8.3
DLCO % predicted	100.2 ± 14.3

NOTE: BMI, body mass index; FEV, forced expiratory volume; FVC, forced vital capacity; TLC, total lung capacity; DLCO, diffusing capacity of the lungs for carbon monoxide.

**Table 2 jcm-11-07276-t002:** The correlation of chest wall and diaphragmatic motion with age.

Ratios		Quiet Breathing				Deep Breathing		
	Age < 30 Years Old (n = 28)	31–50 Years Old (n = 26)	Age > 50 Years Old (n = 20)	*p*	Age < 30 Years Old (n = 28)	31–50 Years Old (n = 26)	Age > 50 Years Old (n = 20)	*p*
**The arcus aortae level**								
Left AP	1.02 ± 0.03	1.02 ± 0.03	1.02 ± 0.03	0.560	1.20 ± 0.14	1.21 ± 0.11	1.22 ± 0.13	0.720
Right AP	1.01 ± 0.02	1.02 ± 0.03	1.01 ± 0.03	0.504	1.22 ± 0.17	1.19 ± 0.10	1.18 ± 0.06	0.844
LR	1.01 ± 0.01	1.01 ± 0.01	1.01 ± 0.02	0.124	1.07 ± 0.04	1.07 ± 0.03	1.08 ± 0.04	0.687
Thoracic Area	1.07 ± 0.05	1.06 ± 0.03	1.07 ± 0.02	0.330	1.44 ± 0.23	1.50 ± 0.22	1.51 ± 0.22	0.384
**The tracheal carina level**								
Left AP	1.02 ± 0.02	1.02 ± 0.03	1.02 ± 0.03	0.672	1.15 ± 0.08	1.15 ± 0.07	1.15 ± 0.07	0.869
Right AP	1.02 ± 0.02	1.01 ± 0.01	1.01 ± 0.02	0.072	1.14 ± 0.08	1.15 ± 0.07	1.14 ± 0.07	0.990
LR	1.01 ± 0.01	1.01 ± 0.01	1.01 ± 0.01	0.211	1.04 ± 0.03	1.04 ± 0.02	1.05 ± 0.03	0.919
Thoracic Area	1.08 ± 0.04	1.07 ± 0.06	1.08 ± 0.05	0.347	1.43 ± 0.23	1.38 ± 0.12	1.36 ± 0.10	0.676
**The liver apex level**								
Left AP	1.01 ± 0.01	1.02 ± 0.03	1.02 ± 0.02	0.247	1.12 ± 0.05	1.13 ± 0.06	1.11 ± 0.03	0.753
Right AP	1.02 ± 0.01	1.01 ± 0.01	1.01 ± 0.02	0.133	1.12 ± 0.05	1.11 ± 0.05	1.10 ± 0.03	0.294
LR	1.01 ± 0.01	1.00 ± 0.01	1.01 ± 0.02	0.546	1.05 ± 0.04	1.04 ± 0.17	1.04 ± 0.02	0.805
Thoracic Area	1.02 ± 0.02	1.01 ± 0.03	1.05 ± 0.06	0.001 *	1.18 ± 0.10	1.20 ± 0.11	1.20 ± 0.09	0.775
**The coronal plane**								
Left CC	1.10 ± 0.05	1.09 ± 0.05	1.10 ± 0.05	0.200	1.33 ± 0.08	1.39 ± 0.12	1.27 ± 0.15	0.081
Right CC	1.10 ± 0.05	1.11 ± 0.07	1.12 ± 0.07	0.750	1.41 ± 0.11	1.47 ± 0.13	1.32 ± 0.16	0.053
Area	1.19 ± 0.11	1.18 ± 0.17	1.21 ± 0.07	0.082	1.88 ± 0.25	1.92 ± 0.33	1.68 ± 0.33	0.150
**The right sagittal plane**								
AOD	1.04 ± 0.02	1.05 ± 0.04	1.05 ± 0.02	0.237	1.15 ± 0.08	1.14 ± 0.07	1.16 ± 0.14	0.852
APD	1.07 ± 0.02	1.09 ± 0.05	1.07 ± 0.09	0.140	1.29 ± 0.10	1.29 ± 0.08	1.30 ± 0.19	0.632
POD	1.09 ± 0.02	1.10 ± 0.05	1.09 ± 0.04	0.896	1.28 ± 0.14	1.33 ± 0.08	1.28 ± 0.13	0.281
Thoracic Area	1.12 ± 0.03	1.13 ± 0.06	1.14 ± 0.04	0.123	1.55 ± 0.19	1.61 ± 0.16	1.60 ± 0.27	0.562

NOTE: * *p* < 0.05. AP, anterior-posterior (AP) distance on the transversal plane; LR, left-right (LR) distance on the transversal plane; CC, the cranial–caudal distance on the coronal plane; AND, the distance between the lung apex and anterior diaphragm on the sagittal plane; APD, the distance between the lung apex and diaphragm apex on the sagittal plane; POD, the distance between the lung apex and posterior diaphragm on the sagittal plane.

**Table 3 jcm-11-07276-t003:** The correlation of chest wall and diaphragmatic motion with BMI.

Ratios	Quiet Breathing			Deep Breathing		
	BMI < 25 (kg/m^2^) (n = 44)	BMI ≥ 25 (kg/m^2^) (n = 30)	*p*	BMI < 25 (kg/m^2^) (n = 44)	BMI ≥ 25 (kg/m^2^) (n = 30)	*p*
**The arcus aortae level**						
Left AP	1.02 ± 0.04	1.02 ± 0.02	0.692	1.17 ± 0.10	1.26 ± 0.14	0.086
Right AP	1.02 ± 0.03	1.01 ± 0.03	0.481	1.18 ± 0.11	1.23 ± 0.13	0.160
LR	1.01 ± 0.01	1.01 ± 0.01	0.093	1.07 ± 0.04	1.08 ± 0.03	0.640
Thoracic Area	1.07 ± 0.04	1.06 ± 0.02	0.895	1.42 ± 0.18	1.58 ± 0.24	0.061
**The tracheal carina level**						
Left AP	1.02 ± 0.03	1.02 ± 0.02	0.567	1.15 ± 0.68	1.16 ± 0.08	0.533
Right AP	1.02 ± 0.02	1.01 ± 0.01	0.271	1.14 ± 0.07	1.14 ± 0.07	0.435
LR	1.01 ± 0.01	1.01 ± 0.01	0.202	1.05 ± 0.03	1.04 ± 0.02	0.763
Thoracic Area	1.08 ± 0.06	1.07 ± 0.02	0.271	1.39 ± 0.19	1.40 ± 0.13	0.876
**The liver apex level**						
Left AP	1.02 ± 0.02	1.04 ± 0.04	0.137	1.12 ± 0.04	1.13 ± 0.07	0.160
Right AP	1.02 ± 0.02	1.01 ± 0.01	0.146	1.11 ± 0.04	1.11 ± 0.05	0.533
LR	1.01 ± 0.01	1.00 ± 0.01	0.134	1.04 ± 0.03	1.05 ± 0.03	0.435
Thoracic Area	1.01 ± 0.01	1.02 ± 0.03	0.792	1.19 ± 0.10	1.20 ± 0.11	0.349
**The coronal plane**						
Left CC	1.10 ± 0.05	1.08 ± 0.04	0.567	1.32 ± 0.12	1.37 ± 0.13	0.275
Right CC	1.11 ± 0.06	1.11 ± 0.06	0.826	1.39 ± 0.13	1.46 ± 0.15	0.533
Thoracic Area	1.21 ± 0.15	1.18 ± 0.08	0.660	1.79 ± 0.29	1.93 ± 0.33	0.533
**The right sagittal plane**						
AOD	1.04 ± 0.03	1.05 ± 0.03	0.378	1.15 ± 0.10	1.15 ± 0.08	0.896
APD	1.08 ± 0.03	1.07 ± 0.07	0.078	1.29 ± 0.12	1.30 ± 0.12	0.275
POD	1.09 ± 0.04	1.09 ± 0.03	0.758	1.28 ± 0.12	1.33 ± 0.12	0.435
Thoracic Area	1.12 ± 0.05	1.13 ± 0.04	0.053	1.56 ± 0.18	1.61 ± 0.22	0.876

NOTE: AP, anterior–posterior (AP) distance on the transversal plane; LR, left–right (LR) distance on the transversal plane; CC, the cranial–caudal distance on the coronal plane; AND, the distance between the lung apex and anterior diaphragm on the sagittal plane; APD, the distance between the lung apex and diaphragm apex on the sagittal plane; POD, the distance between the lung apex and posterior diaphragm on the sagittal plane.

**Table 4 jcm-11-07276-t004:** The correlation of chest wall and diaphragmatic motion with smoking.

Ratios	Quiet Breathing			Deep Breathing		
	Non-Smokers (n = 24)	Smokers (n = 50)	*p*	Non-Smokers (n = 24)	Smokers (n = 50)	*p*
**The arcus aortae level**						
Left AP	1.02 ± 0.02	1.02 ± 0.03	0.474	1.23 ± 0.13	1.18 ± 0.10	0.072
Right AP	1.01 ± 0.03	1.02 ± 0.03	0.853	1.22 ± 0.14	1.19 ± 0.05	0.083
LR	1.01 ± 0.01	1.01 ± 0.01	0.152	1.08 ± 0.03	1.07 ± 0.04	0.212
Thoracic Area	1.06 ± 0.02	1.06 ± 0.04	0.460	1.51 ± 0.23	1.44 ± 0.18	0.152
**The tracheal carina level**						
Left AP	1.01 ± 0.01	1.02 ± 0.03	0.124	1.16 ± 0.08	1.13 ± 0.05	0.065
Right AP	1.01 ± 0.01	1.02 ± 0.02	0.611	1.16 ± 0.07	1.13 ± 0.04	0.052
LR	1.01 ± 0.01	1.01 ± 0.01	0.083	1.05 ± 0.02	1.04 ± 0.03	0.139
Thoracic Area	1.07 ± 0.02	1.08 ± 0.06	0.853	1.42 ± 0.18	1.33 ± 0.12	0.016 *
**The liver apex level**						
Left AP	1.04 ± 0.04	1.02 ± 0.03	0.024 *	1.12 ± 0.06	1.12 ± 0.04	0.853
Right AP	1.01 ± 0.01	1.02 ± 0.02	0.042 *	1.12 ± 0.05	1.11 ± 0.03	0.514
LR	1.00 ± 0.01	1.01 ± 0.01	0.127	1.05 ± 0.03	1.04 ± 0.02	0.432
Thoracic Area	1.03 ± 0.03	1.02 ± 0.05	0.034 *	1.20 ± 0.11	1.19 ± 0.09	0.782
**The coronal plane**						
Left CC	1.09 ± 0.03	1.10 ± 0.05	0.746	1.36 ± 0.11	1.30 ± 0.14	0.072
Right CC	1.12 ± 0.07	1.11 ± 0.06	0.248	1.38 ± 0.12	1.34 ± 0.17	0.098
Thoracic Area	1.20 ± 0.08	1.19 ± 0.15	0.180	1.93 ± 0.28	1.69 ± 0.31	0.003 *
**The right sagittal plane**						
AOD	1.04 ± 0.02	1.05 ± 0.03	0.926	1.15 ± 0.08	1.15 ± 0.12	0.817
APD	1.07 ± 0.08	1.08 ± 0.04	0.268	1.30 ± 0.09	1.29 ± 0.17	0.890
POD	1.10 ± 0.04	1.09 ± 0.04	0.432	1.31 ± 0.12	1.29 ± 0.12	0.548
Thoracic Area	1.13 ± 0.04	1.12 ± 0.05	0.160	1.59 ± 0.19	1.56 ± 0.23	0.488

NOTE: * *p* < 0.05. AP, anterior–posterior (AP) distance on the transversal plane; LR, left–right (LR) distance on the transversal plane; CC, the cranial–caudal distance on the coronal plane; AND, the distance between the lung apex and anterior diaphragm on the sagittal plane; APD, the distance between the lung apex and diaphragm apex on the sagittal plane; POD, the distance between the lung apex and posterior diaphragm on the sagittal plane.

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
