# Peer review of "Characteristics of Diaphragmatic and Chest Wall Motion in People with Normal Pulmonary Function: A Study with Free-Breathing Dynamic MRI"

_jcm, 2022, doi:10.3390/jcm11247276_

Round 1

Reviewer 1 Report

There are many numbers in the paper reflecting a lot of work. However, it is difficult for a clinician to get a clear message. I would suggest to give at least one example of a patient with full values of changes in length and cross-sectional area expressed in cm and squarecm as used in clinical practice. Physiologists might like the quantitative approach but for clinicians who are the typical readers of the journal it is cumbersome to draw conclusions.  The legends are very difficult to understand, and full of orthographic errors e.g. :"The scatter plots present the difference of ratios between different directions on each 131 planes during quiet (a-d) and deep (e-h) breathing. Q-IN/EX ratios, the ratio between insperation 132 and experation during quiet brathing"

Where do the " IPF idiopathic pulmonary fibrosis" in the notes of table 1,2 and 3 come from? In the text it is stated that only healthy subjects had been included. 

I do like the method and suggest that you reflect a little bit in the discussion part what can be done with the method to answer clinically relevant questions, e.g. what happens with a rigid thorax, a pleurisy, a phrenic nerve alteration, ... 

Reviewer 2 Report

Dear Doctor Yang, Dear colleagues,

I read with interest your manuscript as it includes aspects of the physio-pathology of breathing applicable to a research field I specifically care about.

Overall, I do believe your work deserves credit. Extensive English edits are needed to allow full comprehension of your manuscript, though.

A few things to be shared right away are: do you really consider smokers to be healthy subjects? They represent indeed 1/3 of your sample size population. Also: PFTs data seem to not be affected by the portion of smokers included herein; so this triggers another question: how did you quantify their smoking habit? Lastly, in table 4: I believe you need to switch smokers with non-smokers in the headings.

Happy to review your manuscript again as soon as it becomes readier for actual re-submission. Warm regards.

Round 2

Reviewer 2 Report

Dear Doctor Yang & Colleagues,

Congratulation, the manuscript has improved quite a lot.

I do think there is still some work to do, though. See more comments / edits in the attached word file.
